# Energy Consumption Analysis and Efficiency Enhancement in Manufacturing Companies Using Decision Support Method for Dynamic Production Planning (*DSM DPP*) for Solar *PV* Integration

**Simona Skėrė** [1,*] , **Paula Bastida-Molina** [2,3], **Elías Hurtado-Pérez** [2,3] **and Kazimieras Juzėnas** [1]

[1] Faculty of Mechanical Engineering and Design, Kaunas University of Technology, 51424 Kaunas, Lithuania; kazimieras.juzenas@ktu.lt

[2] Instituto Universitario de Investigación en Ingeniería Energética (IIE), Universitat Politècnica de València (UPV), 46022 Valencia, Spain; paubasmo@upv.edu.es (P.B.-M.); ejhurtado@die.upv.es (E.H.-P.)

[3] Departamento de Ingeniería Eléctrica, Universitat Politècnica de València (UPV), 46022 Valencia, Spain

\* Correspondence: simona.bukantaite@ktu.lt; Tel.: 370-630-968-06

**Abstract:** The Industrial Revolution brought major technological progress and the growth of manufacturing, which resulted in significant changes in energy use. However, it also brought about new environmental issues such as increased energy needs, unstable electricity costs, and worsened greenhouse gas effects. Nowadays, it is crucial to analyze energy use to stay competitive. Manufacturers, highly dependent on electricity, can save energy and enhance efficiency by improving production methods. This article presents the findings of a research study conducted on a Lithuanian manufacturing company, aiming to investigate its electricity consumption over a 15-month period from 2022.01 to 2023.03—detailed data about the monthly consumption of the six most powerful machines and their active and standby hours are presented. The total electricity consumption of those matched 173.62 MWh. Employing the Decision Support Method for Dynamic Production Planning (*DSM DPP*), which was previously developed and refined, the study examines the potential for time savings and, subsequently, energy savings, through process reorganization. A detailed three-month production orders observation period demonstrates tangible time savings while using the proposed *DSM DPP*—time savings of approximately 5% can be achieved. Compared to that, production might achieve a 20% productivity increase with advanced technology implementation, so 5% is a great result for an easily adaptable method. Based on this, changes in energy consumption and $CO_2$ emissions due to electricity consumption are calculated and presented knowing that the company uses energy from the grid. Adaptation of the replanning method resulted in a reduction of electricity use by 175 kWh and a reduction of $CO_2$ consumption by 27 kg$CO_2$. With proper production planning, energy and $CO_2$ consumption can be decreased, which is a high priority in today's world.

**Keywords:** energy consumption; $CO_2$ emissions; production planning; decision support method





## 1. Introduction

The Industrial Revolution has had profound implications for energy consumption and economic progress. Europe has witnessed remarkable changes in energy consumption and its environmental implications over the past few decades. It has seen significant growth in energy consumption, with a 40% increase from 1990 to 2020. Nonetheless, the resultant environmental challenges, such as increased energy demand and the amplification of the greenhouse effect, necessitate a comprehensive analysis of energy consumption in manufacturing companies. Thus, this topic gained significant attention in recent times [1]. The European Union has set ambitious targets to further enhance sustainability, including a goal of achieving a net-zero greenhouse gas emissions economy by 2050 [2]. These efforts

reflect Europe's commitment to transitioning towards a greener, more sustainable energy system. Based on this, it is clear that finding an easily accessible everyday way to reduce energy consumption in manufacturing companies could help as well [3].

The production processes encounter challenges in terms of planning and adhering to the established plan on a daily basis. These difficulties often lead to disruptions, energy wastage, time inefficiencies, and resource mismanagement. While high-tech companies specializing in repetitive mass production and employing modern production lines readily embrace Industry 4.0 technologies to optimize their operations [4,5], it is important to note that a significant portion of the economy comprises small and medium-sized enterprises (SMEs). These SMEs adopt a successful business model and play a crucial role in the market due to several factors. These factors include their ability to exhibit versatility in their production capabilities, maintain close customer relationships, and swiftly respond to evolving market demands and individual customer requests [6,7]. Thus, the following studies present a method concentrated specifically on SMEs where production dynamics are unavoidable. This method organizes production in the most efficient possible way and so, as a result, electricity and $CO_2$ consumption are lowered.

This study focuses on a Lithuanian manufacturing company, examining its electricity consumption and investigating the potential for efficiency improvement through process reorganization. The analyzed business is classified as a medium-sized company and most existing businesses belong to this type [8]—SMEs. Specifically for this group, a novel decision support method was created and presented in previous research. The main goal of it is to replan production immediately when that is needed and give the response without waiting for the opinion from an expert. This method does not change or optimize the time or operation itself but changes the sequence of production based on the most suitable scenario, which would lead to the highest profits and minimal time consumption. This method could also be presented as a virtual production manager role [9,10]. Using production time in a more efficient way led to better utilization of machinery and, as a result, less waste of energy while decreasing the number of standby hours. This research presents an investigation of electricity use, during a period of 15 months from January of 2022 until March of 2023. The production reconfiguration plan was made by using data from 3 months—12 December 2022 to 24 March 2023. Using the created Decision Support Method for Dynamic Production Planning (*DSM DPP*) allows the company to achieve time savings and this results in energy savings and $CO_2$ emissions reduction, as defined below.

During the investigation, the company was connected to the Lithuanian electricity grid, which provides higher $CO_2$ emission levels due to its carbon intensity, compared to fully renewable energy. Renewable energy plays a significant role in the context of Industry 4.0. The researched company plans to produce and use its own solar power energy from autumn of 2023. This decision is mostly aimed at the potential for cost savings. Additionally, renewable energy sources produce little to no greenhouse gas emissions, making them an environmentally friendly choice [11]. By adopting renewable energy in their operations, SMEs can reduce their carbon footprint and contribute to mitigating climate change, aligning with sustainability goals and improving their overall environmental performance. The Lithuanian electricity grid mix at the moment is combined from several different sources: coal (2.7%), natural gas (27.7%), hydro (0.5%), biofuels and waste (25.6%), oil (41%), wind and solar (2%), etc. [12]. In 2009, Lithuania eliminated nuclear power by switching to other previously mentioned sources. Figure 1 presents the total energy supply (TES) by source in Lithuania between 1990 and 2021 [12].

This article aims to present possible energy savings and their associated $CO_2$ emissions while replanning production processes with the created method. The existing situation and optimized situation provide differences in energy and $CO_2$ consumption results. However, transferring to fully renewable sources of energy would be additional future research that would follow calculations of a full green energy model.

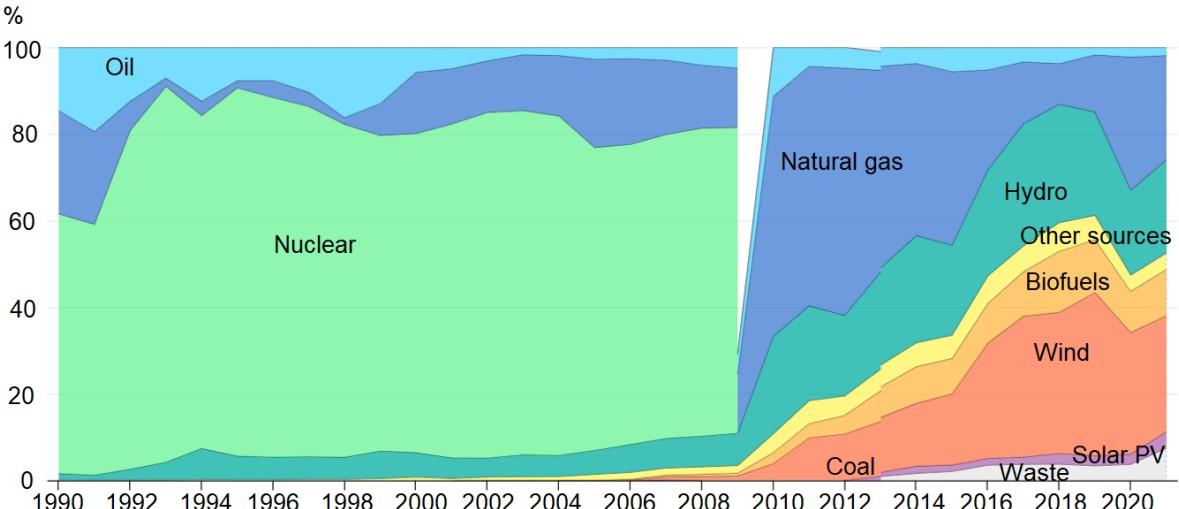

**Figure 1.** Total energy supply (TES) by source, Lithuania, 1990–2021 [12].

The article is presented in five sections. Section 1 is the introduction; Section 2 covers the literature review to present the novelty of the study; Section 3 presents the general methodology for the *DSM DPP*, energy, and CO$_2$ calculations; Section 4 shows the required data for investigation and case study; Section 5 provides the main results and discussion; Section 6 delivers the conclusions.

## 2. Literature Review

This section focuses on the *DSM DPP* compared with existing solutions. Advantages are exposed and thus key points of this method's novelty are presented—fast implementation, focus on employees, and group segmentation of products.

The adoption of enterprise planning systems is accelerating, but compared to large companies, SMEs have a wide range of variables, which makes the implementation of such systems difficult. Deployments usually take 3–9 months [13], but the aim of each company is to achieve its objectives as quickly as possible. The implementation process of the *DSM DPP* is straightforward, requiring only the most important data, thus making the process efficient.

In such production, the variety of products is high and the segmentation of products is essential. The production of a new product requires lead times, and without this, planning is impossible. Therefore, our product will provide times even for unknown products, because the user will simply assign the intended category according to the group technology. The principle of group technology has long been known and its usefulness has been proven [14], but it is rarely used in conjunction with scheduling in the same systems.

At the moment, the transition from Industry 4.0 to Industry 5.0 has become more common. As Industry 4.0 focuses on integrating automation, data exchange, and smart systems into production processes, Industry 5.0 extends its principles by focusing on cooperation between people and machines and the integration of human skills with advanced technologies. By combining the skills of people and machines, Industry 5.0 aims to create flexible and customized production processes. Supporting the well-being of employees becomes a priority, for which it is essential in manufacturing to know the competencies of employees, so that the right task is assigned, which not only generates the most value because it is performed by the most suitable candidate but also creates less tension when an employee is not confident in his or her knowledge [15].

## 3. Methodology

This section presents the general methodology for the whole research and is divided into three subsections: a description of the Decision Support Method for Dynamic Produc-

tion Planning (*DSM DPP*), a mathematical model for energy savings, and $CO_2$ emission reduction calculations.

### 3.1. DSM DPP

The primary objective of the mentioned *DSM DPP* (Decision Support Method for Dynamic Production Planning) is to dynamically adjust the production sequence in quasi-real time. This methodology is specifically designed for SMEs that do not engage in mass production or use innovative production planning strategies. However, such enterprises commonly operate in a rapidly changing environment, where the production of niche or custom-made orders is prevalent [16]. Additionally, these companies often rely on a workforce that places emphasis on employee-centric practices, further adding complexity to the planning process [17]. The most frequently encountered production issues and disruptions in this context include machinery failures, material shortages, quality issues, the introduction of new products, and employee absence [9]. The *DSM DPP* addresses all of these areas and offers prompt solutions, which can range from straightforward replanning to providing decision support information. Production is basically working in the same circle with external disturbances as mentioned. This is presented in Figure 2.

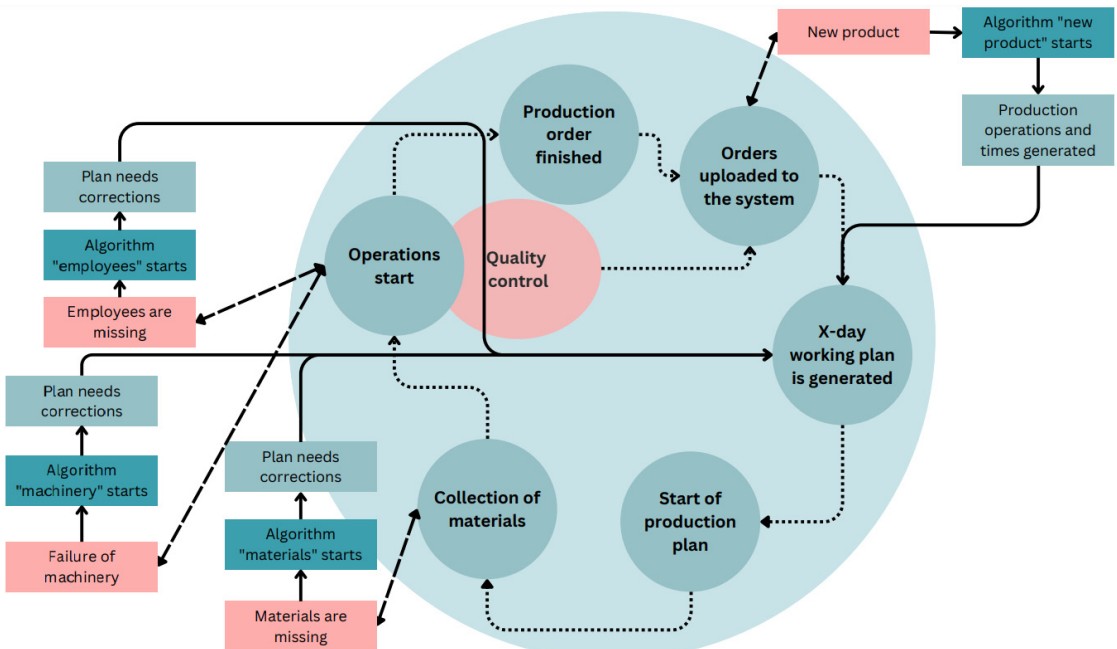

**Figure 2.** Production circle.

The parts in Figure 2 that are marked in red are solved with the created method. Consequently, the overall production time is reduced, as the *DSM DPP* now handles commonly occurring interruptions. The ability to obtain calculated and evaluated solutions within seconds enables companies to proceed with minimal disruptions. Extensive testing of this method in various production companies has demonstrated its adaptability and versatility. By employing this approach, it was observed that the same production time yielded higher production output [10]. In other words, standby time is minimized and converted into active production time. This study aims to emphasize the importance of saving production time and utilizing it to reduce electricity consumption and $CO_2$ emissions, which are described in the following subsections of this section.

The *DSM DPP* works continuously during the whole production process. The main data that are needed should be provided using different easily implemented I4.0 solutions—sensors, Internet of Things (IoT), PID controllers, etc. Multiple data arrays are needed to facilitate the implementation of the proposed method. The successful utilization of this method necessitates consensus on multiple factors and the acquisition of diverse

data inputs. Essential information includes employee skill sets, machine parameters, task priority rankings, and the hourly costs associated with both machinery and employees. By leveraging these data inputs, the method can autonomously make informed decisions and dynamically adjust production processes in response to interruptions. The primary objective of this method is to reduce reliance on human decision-making and enhance operational efficiency. The *DSM DPP* covers a wide range of disruptions that typically require expert judgment. By minimizing human intervention, the *DSM DPP* ensures a more streamlined and efficient decision-making process, thereby reducing the potential for errors and delays. It employs data-driven insights to independently evaluate and respond to production stops, offering timely and optimal solutions. The aim is to maximize production efficiency and minimize disruptions by relying on algorithmic analysis rather than relying solely on human expertise.

*3.2. Electricity Consumption*

Electricity consumption (*E*) quantifies the electricity demanded by the total number of machines in the company during the period under study, normally considering two operating modes: active and standby, as Equation (1) reflects:

$$E = \int_j \sum_m \sum_{i=1}^{i=2} P_{im} \cdot t_{im} \tag{1}$$

where $P_{im}$ represents the active power consumption at each operating mode (*i*), and for each machine (*m*), whereas $t_{im}$ corresponds to the sampling period. Thus, each operating mode (*i*) corresponds to {*i* = 1 (active mode); *i* = 2 (standby mode)}. Finally, index *j* represents the period under study.

Electricity savings ($E_s$) after implementing the proposed *DSM DPP* can be quantified with Equation (2):

$$E_s = E_0 - E_{DSM\ DPP} \tag{2}$$

where $E_0$ and $E_{DSM\ DPP}$ are the consumed electricity in a base scenario and after applying the *DSM DPP*, respectively.

For future analysis of solar photovoltaic (*PV*) integration in the company, different factors should be considered:

Energy Output

The energy generated in output ($E_{out}$) by a solar *PV* system is defined in Equation (3):

$$E_{out} = A \cdot r \cdot H \cdot PR \tag{3}$$

where *A* is the total area of the panel, *r* represents the relationship between the electrical power of the solar panel and its corresponding area, *H* is the average solar radiation per year on tilted panels, and *PR* is the performance ratio (later defined).

The final yield factor ($Y_F$) relates to the total *AC* energy produced by the solar *PV* system for a specific period ($E_{AC}$) with the installed solar *PV* system rated output power ($P_{Rated,\ PV}$), as follows:

$$Y_F = \frac{E_{AC}}{P_{Rated,\ PV}} \tag{4}$$

The reference yield factor ($Y_R$) shows the relationship between the total solar insolation in the plane ($H_{tot}$) with the reference irradiance ($G_{ref}$) of 1 kW/m$^2$:

$$Y_R = \frac{H_{tot}}{G_{Ref}} \tag{5}$$

The performance ratio (*PR*) is defined as the ratio between the final yield to the reference yield.

$$PR = \frac{Y_F}{Y_R} \tag{6}$$

Solar *PV* systems include different losses, which are detailed in Table 1:

**Table 1.** Solar *PV* system losses.

| Description | Abbreviation |
|---|---|
| Loss due to irradiance level | $L_{irr}$ |
| Loss due to temperature | $L_{T^a}$ |
| Module quality loss | $L_{Quality}$ |
| Array mismatch loss | $L_{Mismatch\ loss}$ |
| Ohmic wiring loss | $L_{Ohm}$ |
| Inverter (DC to AC) conversion loss | $L_{Inverter\ DC\text{-}AC}$ |

*3.3. $CO_2$ Emissions Reduction*

According to the type of electricity system that supplies the company in question, $CO_2$ emissions due to this use will vary. Namely, the emissivity of the generation system ($g_{gs}$) will determine its sustainability, since it depends on the percentage that each energy resource represents out of the total, as Equation (7) shows. In it, we can observe how systems that account for a high share of renewable technologies will match lower emissivity values.

$$g_{gs} = \sum_k \frac{E_k}{E} \cdot g_k \tag{7}$$

where *k* is the index for the technologies involved in electricity generation, i.e., solar *PV*, wind, hydropower, coal, and gas, and $g_k$ is the emissivity for each energy resource. Moreover, the weight of each energy technology in the total system depends on the energy that they provide ($E_k$) with respect to the total ($E$).

By coupling this emissivity with the total electricity consumed, $CO_2$ emissions for this use can be obtained.

Furthermore, a reduction in $CO_2$ emissions ($rCO_2$) implies a comparison of two scenarios in which either the emissivity changes (incorporation of more renewable share in the electricity mix, inclusion of solar *PV* for self-consumption, change to an off-grid system, etc.) or the energy consumption, or even both. Equation (8) reflects this:

$$rCO_2 = E_0 \cdot g_0 - E_{DSM\ DPP} \cdot g_{DSM\ DPP} \tag{8}$$

**4. Case Study: Medium-Sized Lithuanian Metal Processing Company**

The research employs a case study approach, selecting a specific Lithuanian manufacturing company as the subject of investigation. This examined company has a workforce of 64 employees, classifying it as a medium-sized establishment. Specializing in the production of furniture components, such as metal tube legs, brackets, and frames for shelves and tables, the company offers a diverse range of over 500 active product article numbers to meet the specific requirements of its customers. In addition to accepting small-scale and individual sample orders, the company provides metal processing services (Table 2). Due to the variable nature of its production, characterized by daily fluctuations, the company places high importance on flexibility, rapid responsiveness, and adaptability. Presently, the company's equipment is maintained by its employees, with no robotic or automated assembly line in place. At the moment there are no specific plans to implement advanced equipment in production.

For a duration of fifteen months, specifically from January 2022 to April 2023, the company provided data on the energy usage of the whole company. Table 3 presents the electricity consumption by month.

**Table 2.** Company details.

| Description | Quantity |
| --- | --- |
| Employees (total) | 64 |
| Employees (administration) | 11 |
| Employees (production) | 53 |
| Active individual article numbers | 524 |
| Products | Carbon steel furniture legs<br>Metal frames<br>Aluminum legs<br>Wooden legs<br>Custom products (shelves, adjustable furniture elements, etc.) |
| Services | CNC turning<br>Welding<br>Woodturning<br>CNC milling<br>Powder coating<br>Bending |

**Table 3.** Current electricity usage by month.

| Year | Month | Electricity (kWh) |
| --- | --- | --- |
| 2022 | January | 20,824 |
| 2022 | February | 15,367 |
| 2022 | March | 20,849 |
| 2022 | April | 17,193 |
| 2022 | May | 20,071 |
| 2022 | June | 16,200 |
| 2022 | July | 10,097 |
| 2022 | August | 17,114 |
| 2022 | September | 16,750 |
| 2022 | October | 13,884 |
| 2022 | November | 18,343 |
| 2022 | December | 14,179 |
| 2023 | January | 17,866 |
| 2023 | February | 13,424 |
| 2023 | March | 15,869 |

The investigation was conducted by the company, which operated in two shifts, each consisting of 8 working hours. It was accepted that an average of 5% of the total electricity consumption was allocated for administrative purposes. The administration consisted of 8 employees, each working 8 h per day. Additionally, it was accepted that 25% of the total electricity consumption could be attributed to low-power production processes, such as lighting, the usage of aerial devices, alarms, cameras, and other similar equipment. The main machinery was listed. These numbers were taken by assuming the quantities of used devices. The company offers services including CNC milling, CNC turning, welding, and powder coating finishing, as indicated in Figure 3, which presents the base assortment.

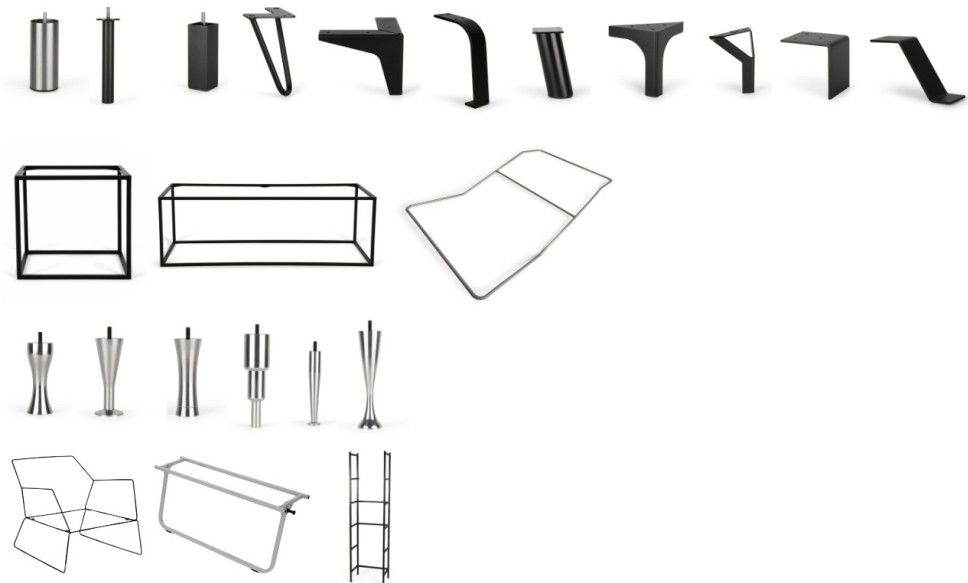

**Figure 3.** From top to bottom: carbon steel furniture legs, frames, aluminum legs, and custom-made products (shelves, adjustable furniture elements).

A total of six key machinery units were found to be responsible for consuming the majority of the electrical energy resources:

- automatic tube cutting machine (M1);
- CNC turning machine (M2);
- painting booth (M3);
- welding machines (four identical pieces) (M4);
- wood turning machine (M5);
- CNC milling machine (M6).

It was imperative to ascertain the power consumption of each machinery unit during both active periods and standby time. A detailed breakdown of this information can be found in Table 4. Consequently, the amount of electricity consumed was disaggregated based on the machinery's utilization within each month. Specifically, the machines were either actively operating or in standby mode during working hours.

**Table 4.** Power of machinery.

| Machinery Code | Active Power (kW) | Standby Power (kW) |
|:---:|:---:|:---:|
| M1 | 4 | 1.5 |
| M2 | 7.5 | 3 |
| M3 | 15 | 4 |
| M4 | 6 | 1.5 |
| M5 | 5 | 1.5 |
| M6 | 7.5 | 3 |

As described, the company is connected to the grid. Lithuania is connected to the wider European electricity grid through several interconnections. The NordBalt submarine HVDC (high-voltage direct current) cable connects Lithuania with Sweden, while the LitPol Link interconnection connects Lithuania with Poland [18]. These interconnections allow for the import and export of electricity, enhancing energy security and enabling the integration of renewable energy sources. Lithuania has been actively transitioning its energy sector to reduce dependence on fossil fuels and increase the share of renewable energy [19,20]. The country has a diverse mix of energy sources, including natural gas, oil, coal, and renewable energy, as presented in Figure 1. The latest statistics from 2022 show the average emissions of Lithuania were 154 g $CO_2$eq/kWh. A total of 78% of energy was

produced from renewable sources, with the main source of energy being wind (36.1%) [21]. Figure 4 presents this specific data from 2022 in more detail. These data will be helpful in calculations of the fourth section, where the main results are presented.

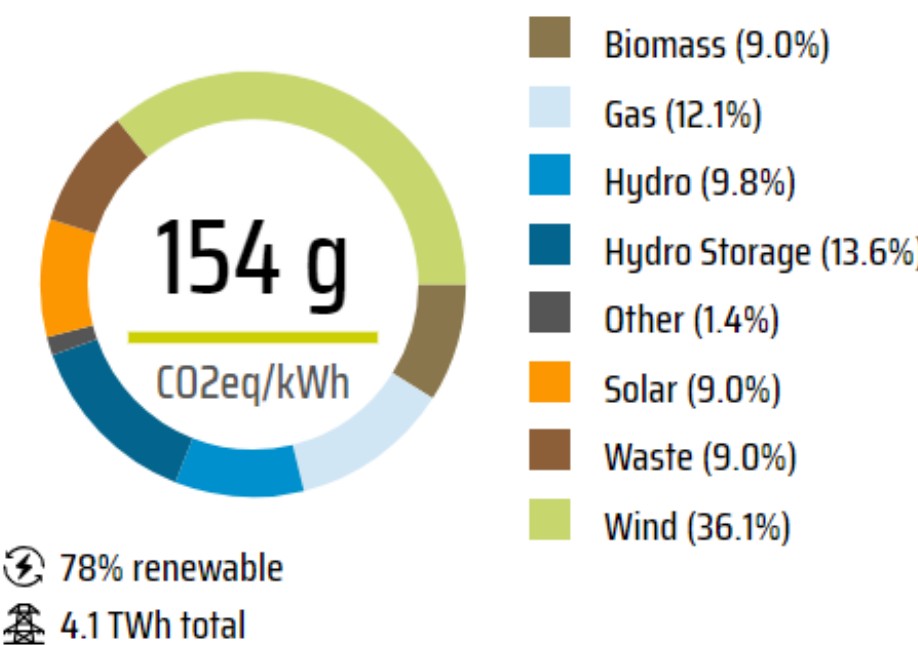

**Figure 4.** Aggregated emissions for 2022 ($CO_2$eq/kWh), Lithuania 2022 [21].

## 5. Results and Discussion

There are two scenarios: the base scenario, in which we have collected data from the company as a fact and the *DSM DPP* is not applied, and the "*DSM DPP* scenario", in which it has been applied and savings of energy are calculated. The *DSM DPP* was implemented to assess the potential for time savings and subsequent energy savings resulting from process rearrangement. The research incorporates a three-month observation period to evaluate the effectiveness of the proposed "*DSM DPP* scenario". Throughout this designated timeframe, all completed production orders were taken into consideration. The aforementioned research spanned from 12 December 2022 to 24 March 2023, encompassing a total of 72 working days conducted in two shifts. A grand total of 491 production orders were executed, and Figure 5 depicts a selected portion of the collected data. The historical production order data were provided by the company and subsequently extracted and organized to obtain the required information for testing with *DSM DPP*. The presented information is the base scenario data.

| Company | Production order no. | Product no. | Quantity | Days | Date | Price, eur | Price-material, | delivery partly | Payment type | Operations | 1 | 2 | 3 | 4 | 5 | 6 | 7 | 8 | 9 | 10 | 11 | 12 | 13 | Current stockmin / Rating | delivery reliability | New | Delivery product | Complexity production | Reject rate | Subcontract orders | subcontractors | Materials |
|---|---|---|---|---|---|---|---|---|---|---|---|---|---|---|---|---|---|---|---|---|---|---|---|---|---|---|---|---|---|---|---|---|
| | | | | | | | | | | | | | | | | | | | | | | | | Time of operation (1 piece), min | | | | | | | | |
| C | 3 | p1 | 1 | 20 | 2022–11–12 | 186 | 86 | 0 | 60 | 10, 5, 11, 13 | 0 | 0 | 0 | 0 | 30 | 0 | 0 | 0 | 0 | 120 | 60 | 0 | 12 | 0.9 | 20 | 0 | 1 | 0.8 | 0.05 | 0 | 0 | 1, 2 |
| E | 9 | p1 | 1 | 20 | 2022–11–17 | 186 | 86 | 0 | −1 | 10, 5, 11, 13 | 0 | 0 | 0 | 0 | 30 | 0 | 0 | 0 | 0 | 120 | 60 | 0 | 12 | 0.65 | 0 | 0 | 2 | 0.8 | 0.05 | 0 | 0 | 1, 2 |
| G | 10 | p1 | 1 | 20 | 2022–11–17 | 186 | 86 | 0 | 30 | 10, 5, 11, 13 | 0 | 0 | 0 | 0 | 30 | 0 | 0 | 0 | 0 | 120 | 60 | 0 | 12 | 0.8 | 15 | 0 | 1 | 0.8 | 0.05 | 0 | 0 | 1, 2 |
| C | 26 | p1 | 1 | 20 | 2022–11–23 | 186 | 86 | 0 | 60 | 10, 5, 11, 13 | 0 | 0 | 0 | 0 | 30 | 0 | 0 | 0 | 0 | 120 | 60 | 0 | 12 | 0.9 | 20 | 0 | 1 | 0.8 | 0.05 | 0 | 0 | 1, 2 |
| C | 28 | p1 | 1 | 20 | 2022–11–24 | 186 | 86 | 0 | 60 | 10, 5, 11, 13 | 0 | 0 | 0 | 0 | 30 | 0 | 0 | 0 | 0 | 120 | 60 | 0 | 12 | 0.9 | 20 | 0 | 1 | 0.8 | 0.05 | 0 | 0 | 1, 2 |
| C | 26 | p1 | 1 | 20 | 2022–11–23 | 186 | 86 | 0 | 60 | 10, 5, 11, 13 | 0 | 0 | 0 | 0 | 30 | 0 | 0 | 0 | 0 | 120 | 60 | 0 | 12 | 0.9 | 20 | 0 | 1 | 0.8 | 0.05 | 0 | 0 | 1, 2 |
| C | 28 | p1 | 1 | 20 | 2022–11–24 | 186 | 86 | 0 | 60 | 10, 5, 11, 13 | 0 | 0 | 0 | 0 | 30 | 0 | 0 | 0 | 0 | 120 | 60 | 0 | 12 | 0.9 | 20 | 0 | 1 | 0.8 | 0.05 | 0 | 0 | 1, 2 |
| F | 29 | p1 | 1 | 20 | 2022–11–24 | 186 | 86 | 0 | −1 | 10, 5, 11, 13 | 0 | 0 | 0 | 0 | 30 | 0 | 0 | 0 | 0 | 120 | 60 | 0 | 12 | 0.7 | 0 | 0 | 2 | 0.8 | 0.05 | 0 | 29 | 1, 2 |
| L | 132 | p1 | 1 | 20 | 2022–12–16 | 186 | 86 | 0 | −1 | 10, 5, 11, 13 | 0 | 0 | 0 | 0 | 30 | 0 | 0 | 0 | 0 | 120 | 60 | 0 | 12 | 0.75 | 15 | 0 | 2 | 0.8 | 0.05 | 0 | 0 | 1, 2 |
| L | 132 | p1 | 1 | 20 | 2022–12–16 | 186 | 86 | 0 | −1 | 10, 5, 11, 13 | 0 | 0 | 0 | 0 | 30 | 0 | 0 | 0 | 0 | 120 | 60 | 0 | 12 | 0.75 | 15 | 0 | 2 | 0.8 | 0.05 | 0 | 0 | 1, 2 |
| M | 133 | p1 | 1 | 20 | 2022–12–22 | 186 | 86 | 0 | −1 | 10, 5, 11, 13 | 0 | 0 | 0 | 0 | 30 | 0 | 0 | 0 | 0 | 120 | 60 | 0 | 12 | 0.8 | 40 | 0 | 2 | 0.8 | 0.05 | 0 | 0 | 1, 2 |
| M | 133 | p1 | 1 | 20 | 2022–12–22 | 186 | 86 | 0 | −1 | 10, 5, 11, 13 | 0 | 0 | 0 | 0 | 30 | 0 | 0 | 0 | 0 | 120 | 60 | 0 | 12 | 0.8 | 40 | 0 | 2 | 0.8 | 0.05 | 0 | 0 | 1, 2 |
| M | 133 | p1 | 1 | 20 | 2022–12–22 | 186 | 86 | 0 | −1 | 10, 5, 11, 13 | 0 | 0 | 0 | 0 | 30 | 0 | 0 | 0 | 0 | 120 | 60 | 0 | 12 | 0.8 | 40 | 0 | 2 | 0.8 | 0.05 | 0 | 0 | 1, 2 |
| G | 86 | p1 | 1 | 20 | 2022–12–01 | 186 | 86 | 0 | 30 | 10, 5, 11, 13 | 0 | 0 | 0 | 0 | 30 | 0 | 0 | 0 | 0 | 120 | 60 | 0 | 12 | 0.8 | 15 | 0 | 1 | 0.8 | 0.05 | 0 | 0 | 1, 2 |
| H | 96 | p1 | 1 | 20 | 2022–12–01 | 186 | 86 | 0 | 30 | 10, 5, 11, 13 | 0 | 0 | 0 | 0 | 30 | 0 | 0 | 0 | 0 | 120 | 60 | 0 | 12 | 0.8 | 50 | 0 | 1 | 0.8 | 0.05 | 0 | 0 | 1, 2 |

**Figure 5.** Fragment of collected data.

It is seen from Figure 3 that the quantity of orders varies for even one piece. Such fluctuations in production volume result in varying energy consumption patterns on a monthly basis, as well as differential electricity usage for individual machines. The specifics regarding the number of working days in each investigated month, along with the corresponding total working hours, are presented in Table 5.

**Table 5.** Investigated period by working days and hours.

| Year | Month | Working Days | Working Hours (of 2 Shifts) |
|---|---|---|---|
| 2022 | January | 21 | 336 |
| 2022 | February | 19 | 304 |
| 2022 | March | 22 | 352 |
| 2022 | April | 20 | 320 |
| 2022 | May | 22 | 352 |
| 2022 | June | 21 | 336 |
| 2022 | July | 12 | 192 |
| 2022 | August | 20 | 320 |
| 2022 | September | 20 | 320 |
| 2022 | October | 17 | 272 |
| 2022 | November | 20 | 320 |
| 2022 | December | 15 | 240 |
| 2023 | January | 22 | 352 |
| 2023 | February | 19 | 304 |
| 2023 | March | 21 | 336 |

In light of the aforementioned information, Figures 6–11 are derived to delineate the temporal aspects of electricity consumption for each machine. These chart representations provide insight into the duration during which each machine operated in both active and standby modes. The figures are divided for each mentioned machine and the blue color presents active hours in each month of observation. Active hours are when the machinery is fully working. The orange color is a representation of standby hours, which are when machinery is not working at full power.

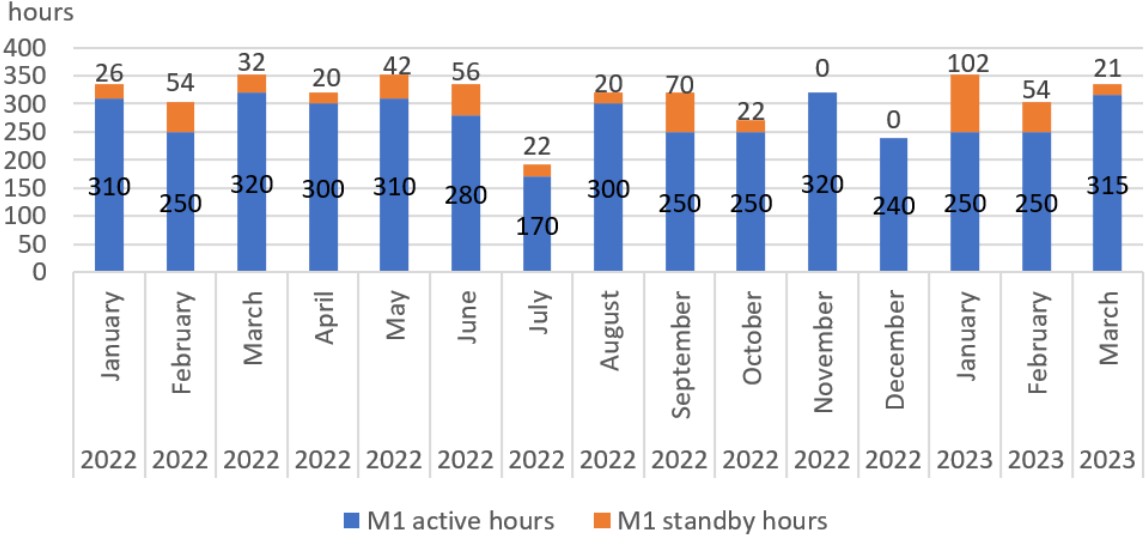

**Figure 6.** M1 machine, active and standby hours, period of 2022.01–2023.03 (base scenario).

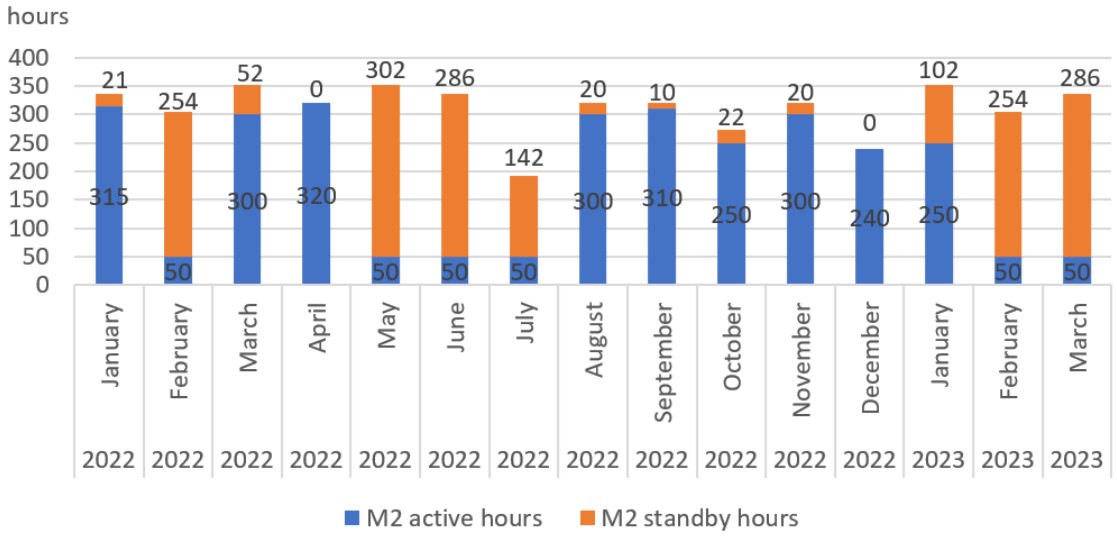

**Figure 7.** M2 machine, active and standby hours, period of 2022.01–2023.03 (base scenario).

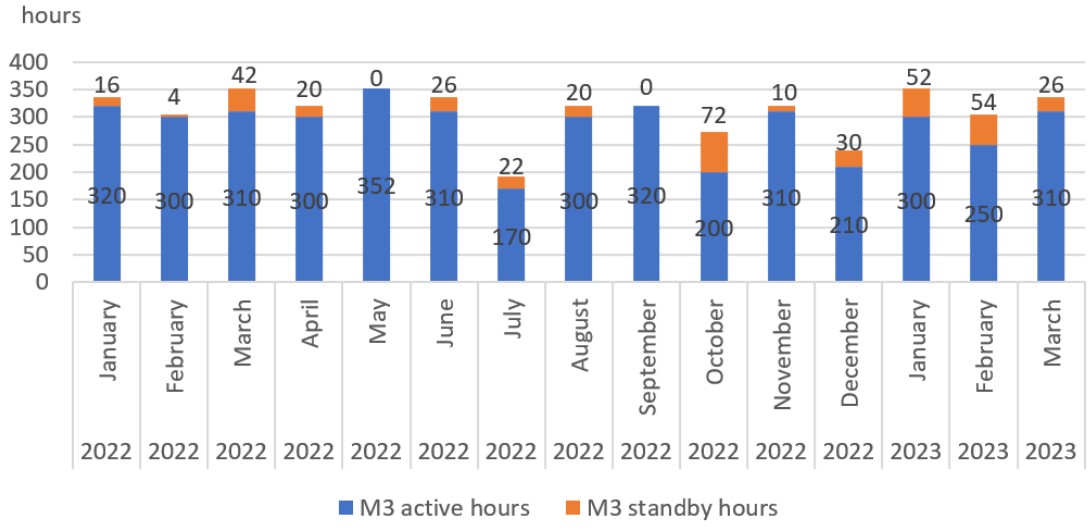

**Figure 8.** M3 machine, active and standby hours, period of 2022.01–2023.03 (base scenario).

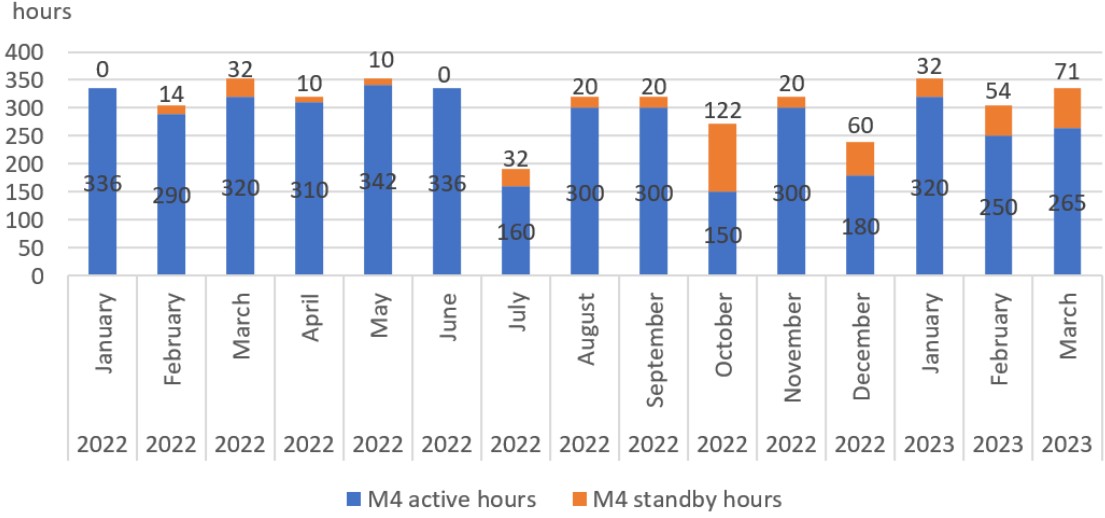

**Figure 9.** M4 machine, active and standby hours, period of 2022.01–2023.03 (base scenario).

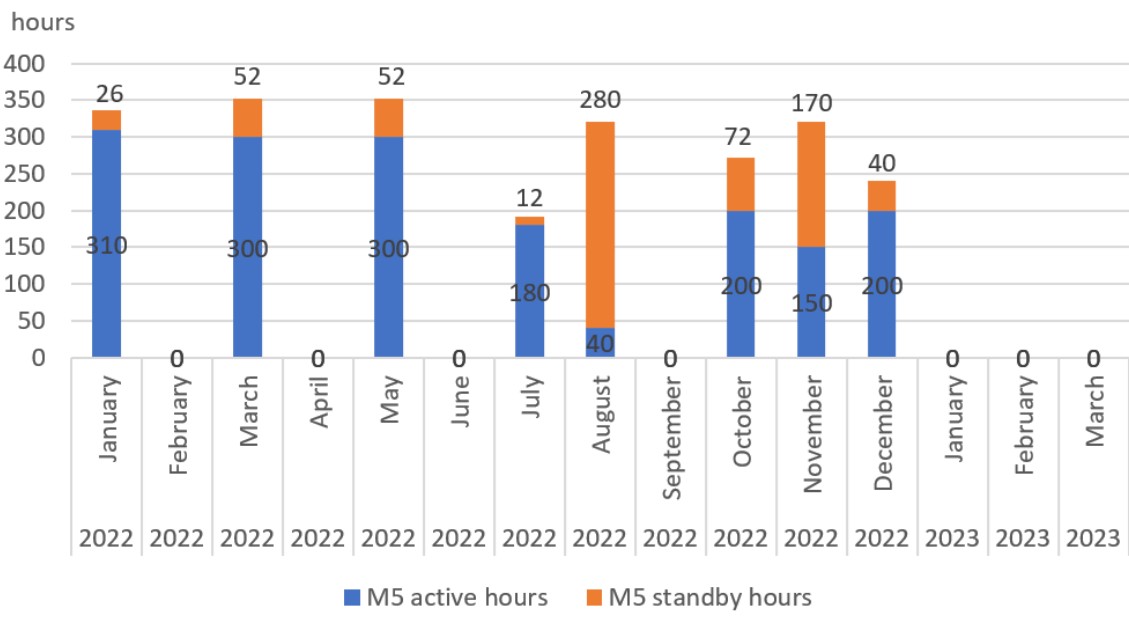

**Figure 10.** M5 machine, active and standby hours, period of 2022.01–2023.03 (base scenario).

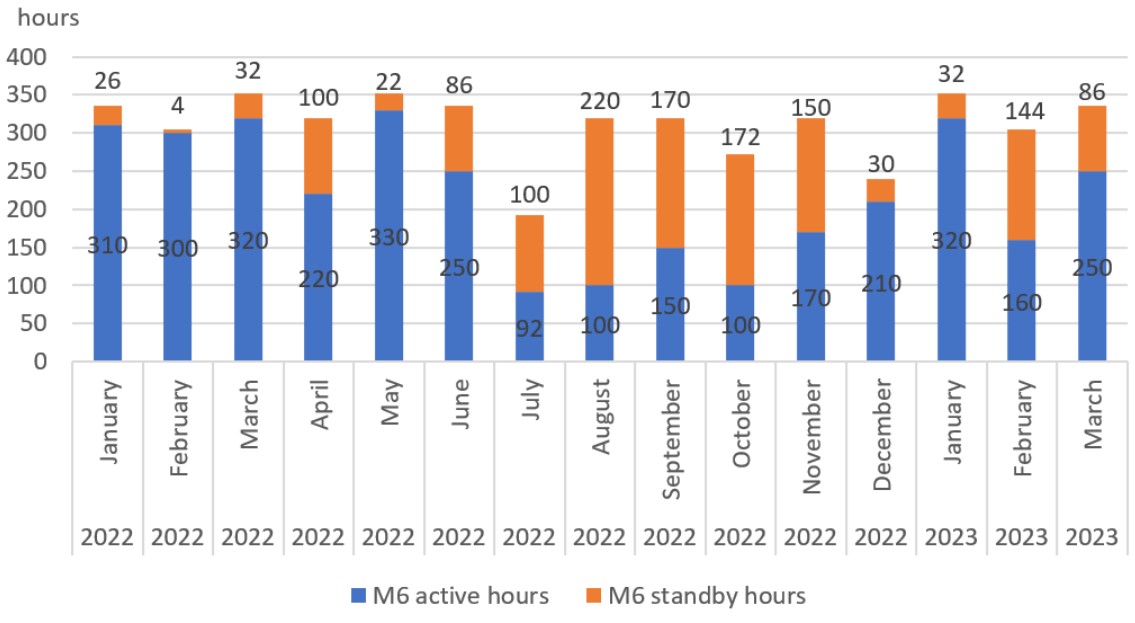

**Figure 11.** M6 machine, active and standby hours, period of 2022.01–2023.03 (base scenario).

This part of the results presented the base scenario and its data from the company.

By coupling this information with the power specifications of each machine, as outlined in Table 2, the requisite data for subsequent energy consumption for the base scenario can be obtained, as Figures 12–17 show. The total electricity consumption due to the six machines for the period 2022.01 to 2023.03 was 173.62 MWh. From them, M3 accounted for the highest total electricity consumption (37.7%), while M5 was the lowest (5.5%). On the other side, 92% of the whole demand corresponded to active electricity consumption and the other 8% to standby electricity consumption. In this regard, M3 stands out from the others with a standby consumption of 30.8%, and M4 was the one with the lowest standby consumption (4.3%).

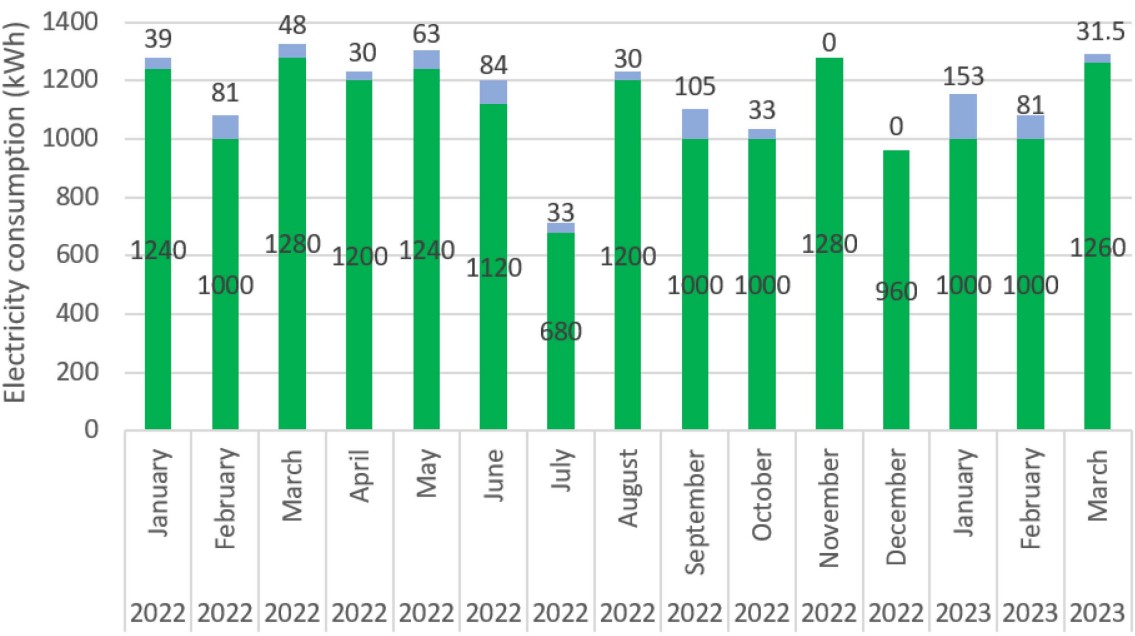

**Figure 12.** M1 machine, active and standby electricity consumption, period of 2022.01–2023.03 (base scenario).

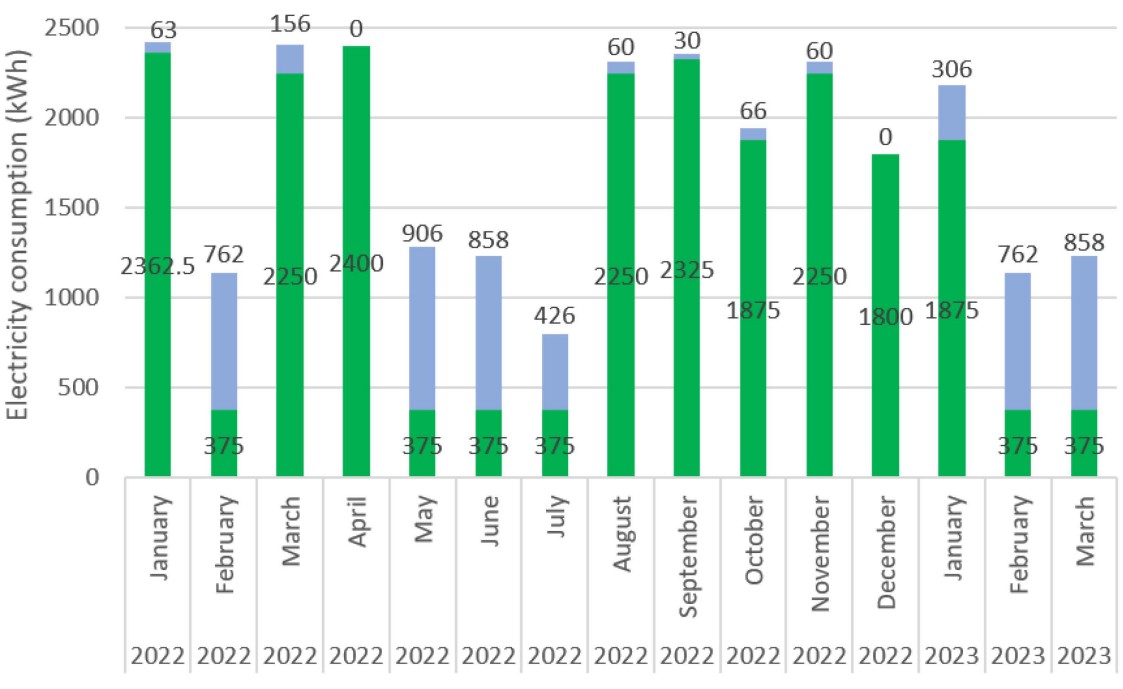

**Figure 13.** M2 machine, active and standby electricity consumption, period of 2022.01–2023.03 (base scenario).

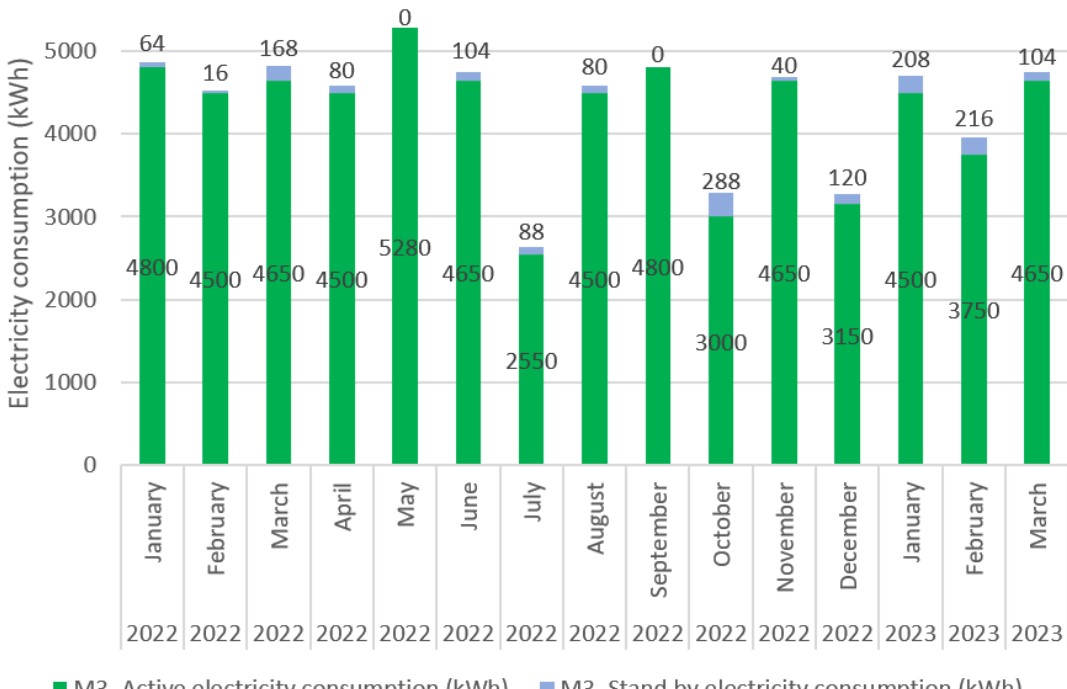

**Figure 14.** M3 machine, active and standby electricity consumption, period of 2022.01–2023.03 (base scenario).

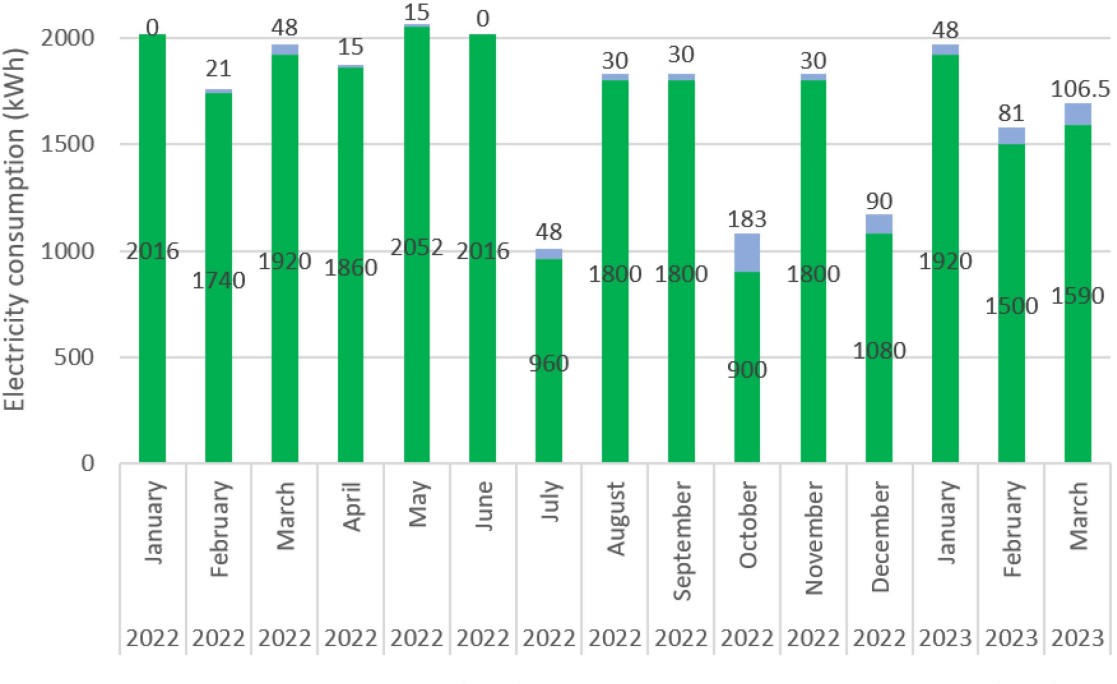

**Figure 15.** M4 machine, active and standby electricity consumption, period of 2022.01–2023.03 (base scenario).

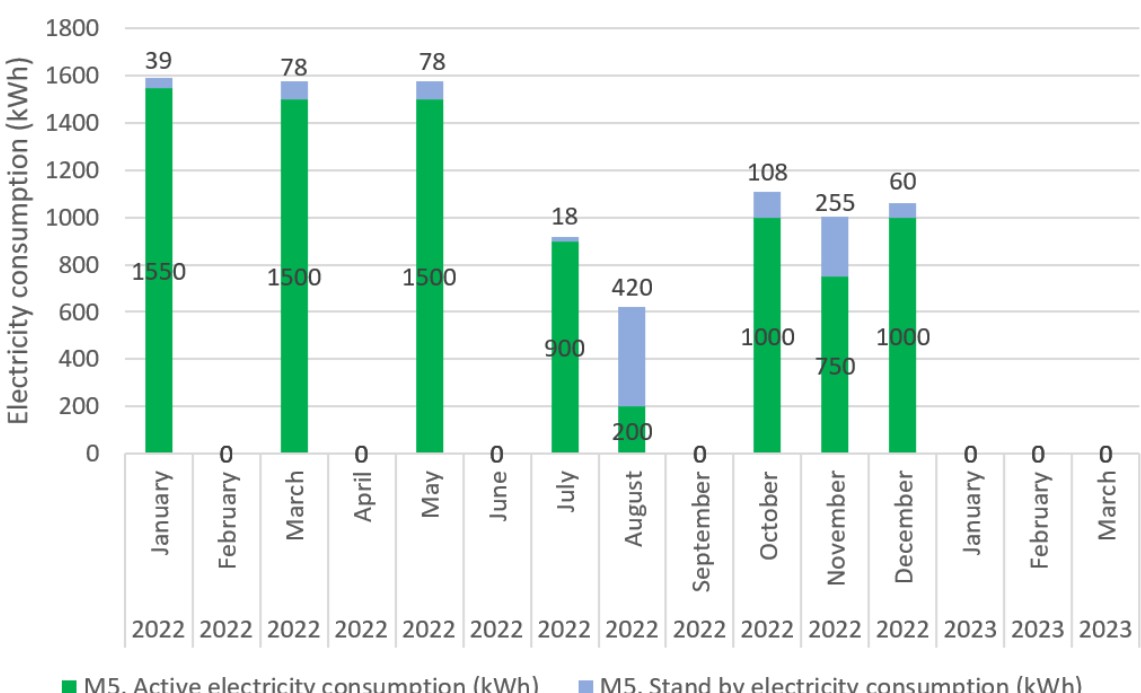

**Figure 16.** M5 machine, active and standby electricity consumption, period of 2022.01–2023.03 (base scenario).

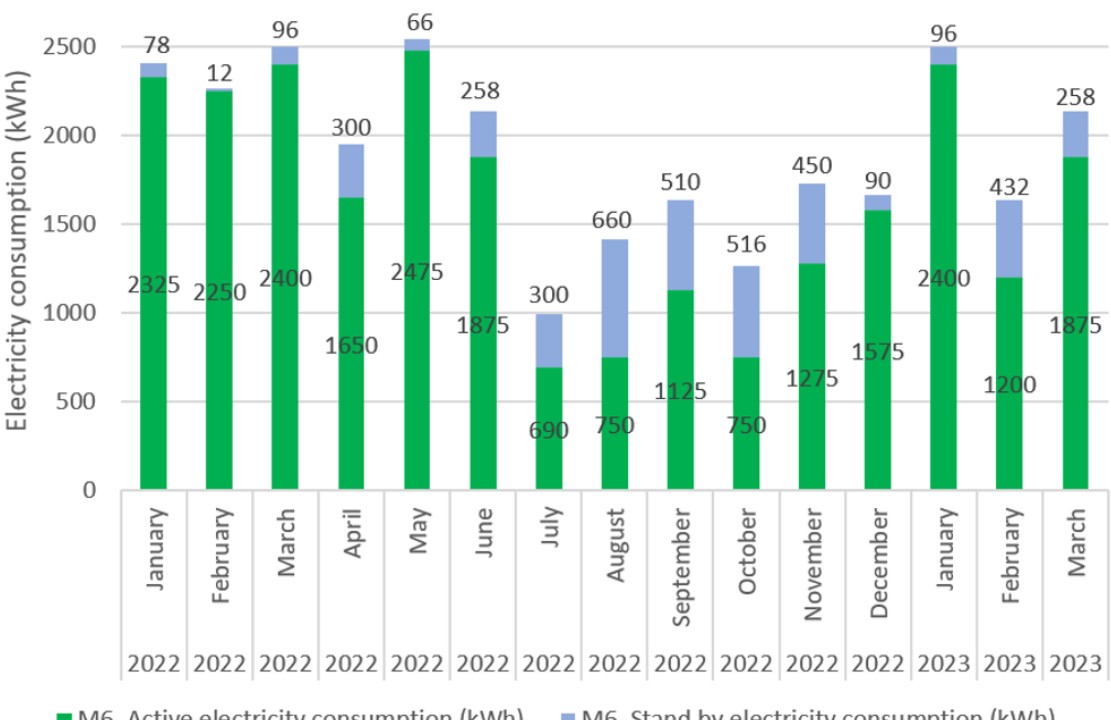

**Figure 17.** M6 machine, active and standby electricity consumption, period of 2022.01–2023.03 (base scenario).

Preliminary findings indicate that the implementation of the *DSM DPP* enables notable time savings in the manufacturing processes of the selected Lithuanian company. These time savings, achieved through optimized process reorganization, offer the potential for corresponding energy savings. The observed three-month period demonstrates promising outcomes, suggesting that the proposed approach has the potential to improve energy effi-

ciency in manufacturing operations. As mentioned earlier, the total period of examination was 72 working days. After the DSM DPO adaptation, all of these orders were filled in 68 days, giving an average of 5% time savings. This is shown in Figure 18.

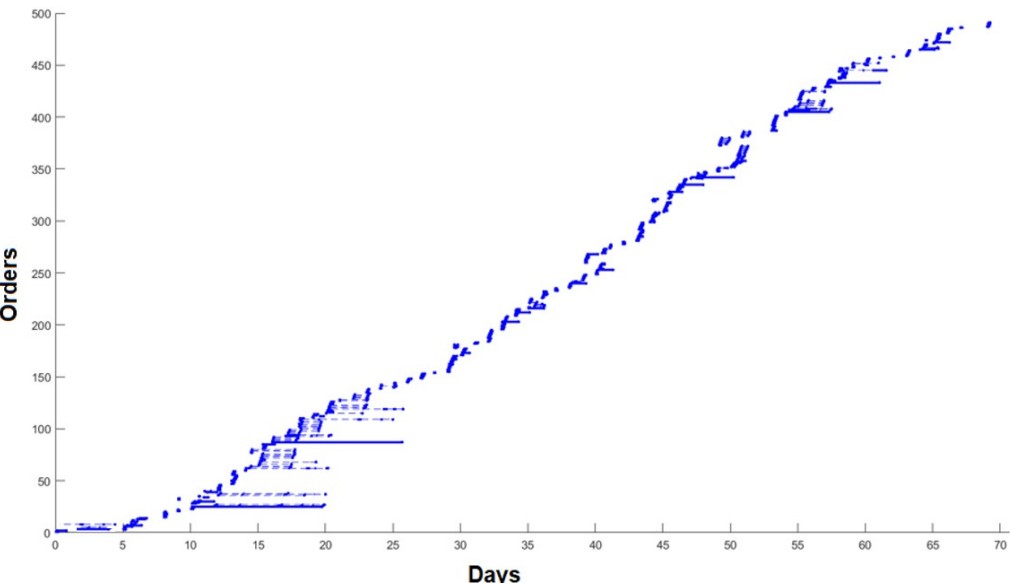

**Figure 18.** Sequence of orders after *DSM DPP* adaptation.

As presented, total savings during this observed period was 4 working days which can be converted to 64 working hours based on the fact that production was working in two shifts of 8 h. As the time of operations has not been changed, these hours are only saved out of standby time. Production operations stay the same since this method does not influence the process itself. It only optimizes the sequence of processes and divides them into several groups.

The observation was made between the middle of December to almost the end of March. Thus, the overall time of active and standby hours of machines M1–M6 are shown from data for half of December 2022 and January, February, and March 2023. This is given in Figure 19.

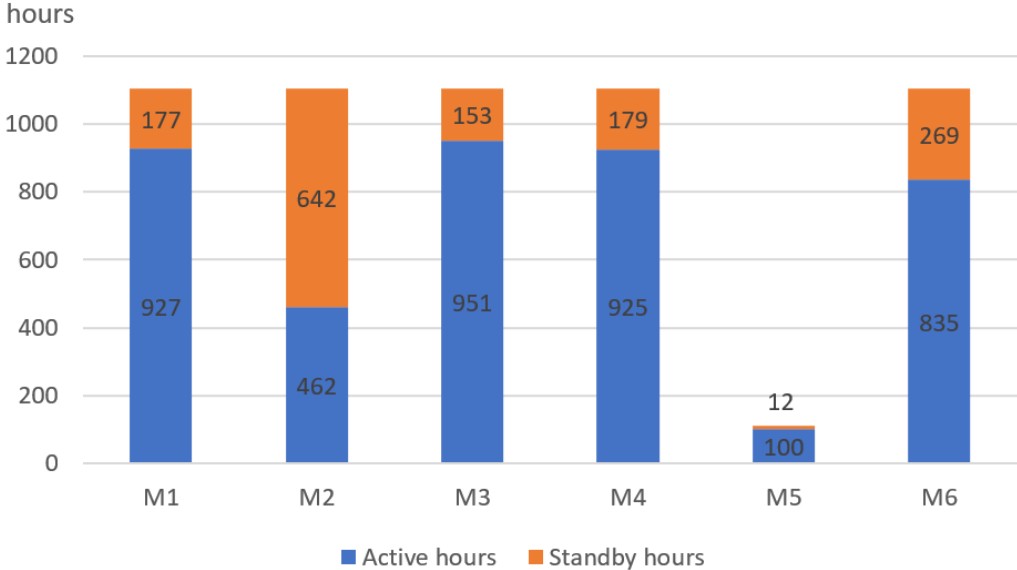

**Figure 19.** M1-M6 machines, active and standby hours during the observed period (*DSM DPP* scenario).

This means that standby time, which in total was 1432 h, was reduced by 64 h, which is 4.5%. The reduced time displacement is presented in Figure 20.

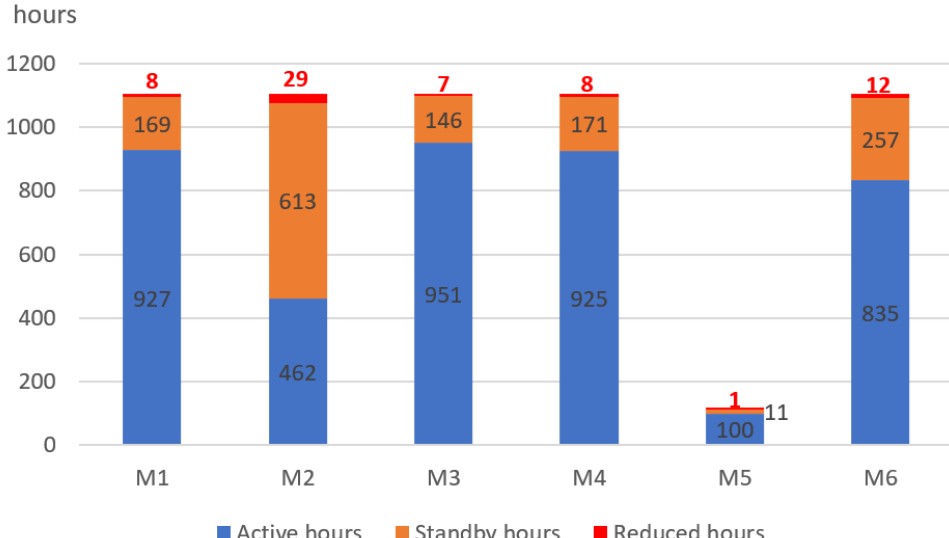

**Figure 20.** M1-M6 machines, active, standby, and reduced hours during the observed period (*DSM DPP* scenario).

The reduction of standby hours for the three-month period under study resulted in a decrease in electricity consumption, as Figures 21 and 22 represent. Namely, the total electricity consumption was reduced by 175 kWh after applying the *DSM DPP*. In this regard, the highest electricity reduction corresponded to M2, with a decrease of 86.7 kWh. This situation matched two circumstances: this machine has the highest standby hours decrease after the application of the *DSM DPP* (Figure 21), and it has a relatively high standby power consumption (Table 3).

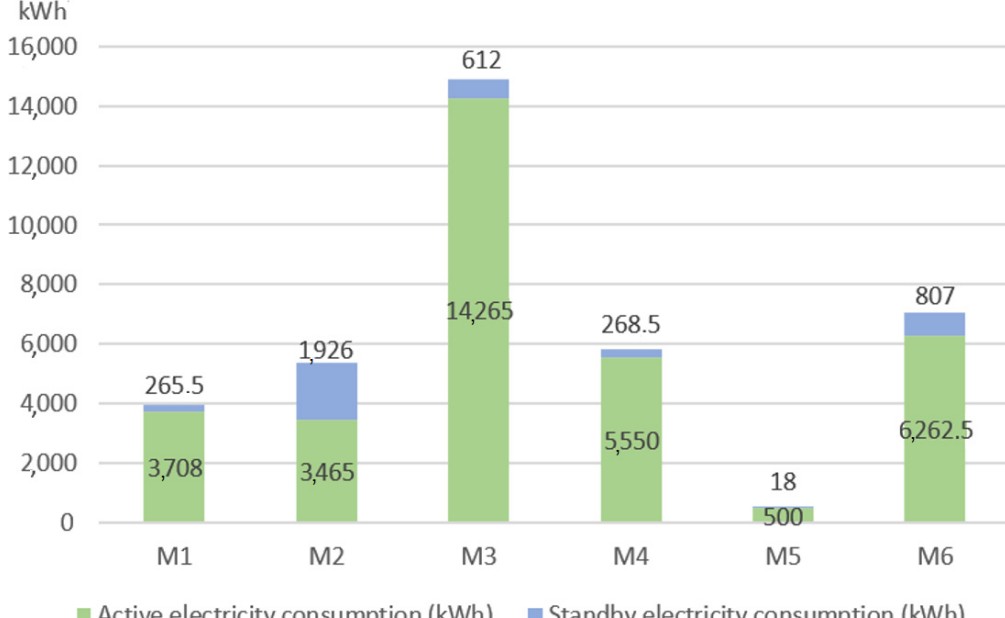

**Figure 21.** M1-M6 machines, active and standby electricity consumption during the observed period (base scenario).

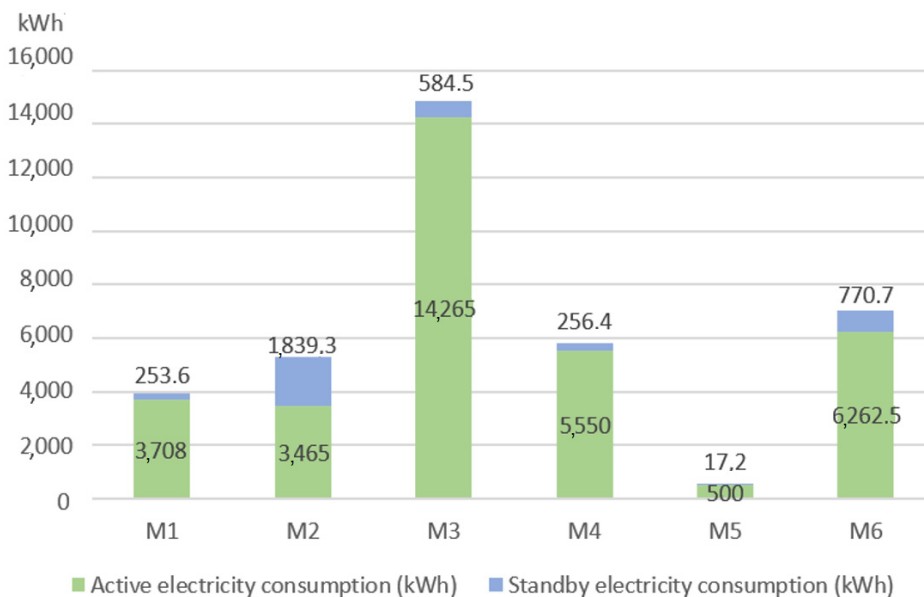

**Figure 22.** M1-M6 machines, active and standby electricity consumption during the observed period (*DSM DPP* scenario).

Furthermore, the company under study is connected to the Lithuanian electricity grid, with an emissivity of 154 $gCO_2$/kWh [21]. Moreover, the company does not currently have any self-consumption renewable system. Thus, these above-mentioned electricity savings lead to a decrease in the $CO_2$ emissions of the company due to electricity consumption. Specifically, the $CO_2$ emission reduction due to electricity consumption of the six machines after applying our method corresponded to 27 $kgCO_2$, compared to the base scenario. M3 accounts for the highest emissions value (40%), while M5 is the lowest (1%), matching the energy consumption rates (Figure 23).

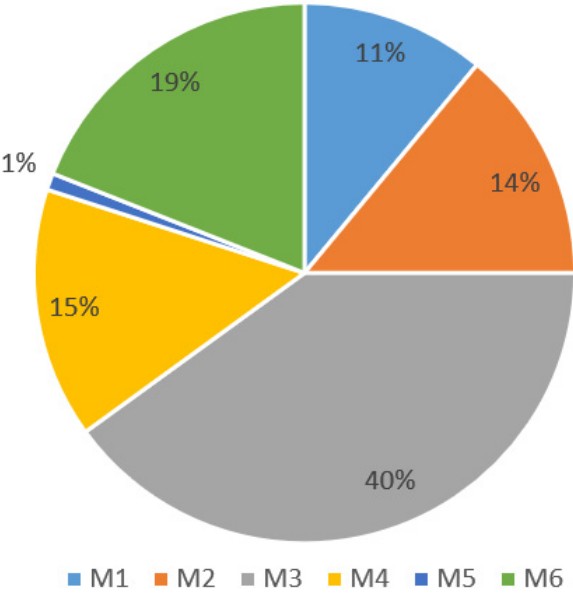

**Figure 23.** $CO_2$ emission values for machines M1–M6.

The results highlight the significance of process reorganization as a means to enhance energy efficiency in manufacturing companies. By optimizing production processes, companies can achieve time savings, thereby reducing electricity consumption and minimizing their environmental impact.

The application of the *DSM DPP* method in the case study demonstrates its potential as a decision support tool for dynamic production planning, facilitating energy savings in manufacturing operations. The manufacturing industry relies heavily on electricity for powering machinery, equipment, and assembly lines. Industries such as automotive, electronics, textiles, and food processing require significant electricity consumption. The presented method can be easily adapted to different types of manufacturing industries.

## 6. Conclusions

By employing the *DSM DPP*, manufacturing companies can enhance their competitiveness while simultaneously reducing their ecological footprint. The investigated case presented time savings of 5% (from 72 to 68 working days after the implementation of the *DSM DPP*). In total, 491 production orders were examined, and information about each of these orders was collected from the metal processing company.

Energy savings of 175 kWh were achieved after applying the proposed method, with a reduction of 27 $kgCO_2$ due to electricity consumption. Although these results show the importance of process planning with the *DSM DPP*, the authors consider that further and stronger environmental measurements should be studied for energy sustainability in the company. Specifically, future research will focus on renewable energy systems design for the company, considering grid-connected and off-grid options.

These findings contribute to the broader discourse on energy consumption in the manufacturing sector and inform strategies for achieving sustainable and resource-efficient operations.

**Author Contributions:** Conceptualization, S.S.; methodology, S.S. and P.B.-M.; validation, S.S., K.J. and E.H.-P.; formal analysis, S.S.; investigation, S.S. and P.B.-M.; resources, S.S.; data curation, S.S.; writing—original draft preparation, S.S.; writing—review and editing, S.S., P.B.-M., K.J. and E.H.-P.; visualization, S.S. and E.H.-P.; supervision, P.B.-M., K.J. and E.H.-P.; project administration, S.S., P.B.-M. and E.H.-P. All authors have read and agreed to the published version of the manuscript.

**Funding:** This research was prepared during the ERASMUS+ funded traineeship.

**Data Availability Statement:** Not applicable.

**Acknowledgments:** We are grateful to the management of the investigated company for providing access to their data and giving us the rights to use it for this research.

**Conflicts of Interest:** The authors declare no conflict of interest.

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
