# Peer review of "Energy Consumption Analysis and Efficiency Enhancement in Manufacturing Companies Using Decision Support Method for Dynamic Production Planning (DSM DPP) for Solar PV Integration"

_machines, doi:10.3390/machines11100939_

Round 1

Reviewer 1 Report

This paper studies the conducted on a Lithuanian manufacturing company, which has potential application value in engineering. In order to meet the requirements of high-quality publication of the journal (Machines), it is recommended to consider the following suggestions.

1) The lenght of title is too long.

2) Except of "approximately 5% of time", there is no key quantitative data in the abstract.

3) The background section of the abstract is too long and needs to be simplified

4) There are too many keywords. Need to refine.

5) In line 32, please check the text of "DSM DPP."

6) I think the Abbr. of "DSM DPP" in line 107 are not good.

7) The DPI of Fig.2 is not good.

8) The mehtod proposed in this paper needs to be compared with the previous literature, otherwise it cannot reflect innovation.

9) It is best to have a separate discussion section.

10) There are too many conclusions that need to be refined.

11) It should also be noted that the number of references cited is insufficient.

Author Response

We thank the reviewers for their constructive comments. We have addressed the referees’ comments and modified the paper accordingly. Our detailed answers follow. Please note that the reviewers' comments are in bold while our answers are not. Furthermore, additions to the revised manuscript are indicated in red.

Itemized reply to reviewer 1

Dear reviewer,

Thank you for such a constructive review.

Below, we answer your comments one by one:

1) The lenght of title is too long.

Thanks for this recommendation. The title was changed and a bit shortened.

2) Except of "approximately 5% of time", there is no key quantitative data in the abstract.

We appreciate this observation and have added more information from our investigation in the abstract.

3) The background section of the abstract is too long and needs to be simplified.

Thank you for the observation. We have simplified and shortened the background.

4) There are too many keywords. Need to refine.

Thanks for the comment. We reduced the number of keywords.

5) In line 32, please check the text of "DSM DPP."

Thanks for the advice. We have corrected it.

6) I think the Abbr. of "DSM DPP" in line 107 are not good.

 Thanks. We have corrected it.

7) The DPI of Fig.2 is not good.

Thank you. We corrected it in the revised manuscript.

 8) The mehtod proposed in this paper needs to be compared with the previous literature, otherwise it cannot reflect innovation.

Thanks for your recommendation. We have added Literature review section in which novelty is described.

9) It is best to have a separate discussion section.

We appreciate this comment. Due to the requirements of the journal and the structure of the paper, we were not able to make a separate discussion section.

10) There are too many conclusions that need to be refined.

Thanks for this observation. We have refined the conclusions section.

11) It should also be noted that the number of references cited is insufficient.

Thank you for this comment. We have expanded the number of references.

Reviewer 2 Report

the paper well presented 

in the section Electricity consumption you need to add the following factor energy output, final and reference yield, Performance Ratio,  System total losses...

the calculation of all factor add to your paper more scientific qualitity

Author Response

Dear reviewer,

Thank you for your comments.

We have added several calculations: Energy output, Final Yield, Reference Yield, Perfomance Ratio, System total losses. 

We have addressed the referees’ comments and modified the paper accordingly. 

Reviewer 3 Report

The paper is interesting, but the subject of the paper does not fit this journal.

The article contains a lot of language errors and incorrect verb forms.

Author Response

Dear reviewer,

Thank you for your comments.

We certainly think that our paper fits the scope of the journal, especially the research topic “advanced manufacturing”, since we are providing a Decision Support Method for Dynamic Production. Using this method, we analyze Energy Consumption and Efficiency Enhancement in Manufacturing Companies that implement this method for solar PV integration, which fits the scope of the Special Issue “Sustainable renewable generation systems”. To support this last topic, we have included solar PV analysis factors in “Electricity consumption” section. Hence, future integration of solar PV in companies could be assessed.

Furthermore, we have deeply reviewed the paper to fix language errors and incorrect verb forms. A native language speaker has checked also the manuscript.

Reviewer 4 Report

This paper probes the electricity consumption trends of a Lithuanian manufacturer, leveraging the DSM DPP method to unveil potential efficiencies. Through restructured production planning, it reveals potential reductions in energy use and CO2 emissions, emphasizing modern environmental imperatives.

The results are good, but the visual presentation lacks order: figures are not numbered ascending (e.g.:3,4, 6 (wrong), ..9, then 5, 10, and so on. The position of the number on figures can be improved. Fig.5 can be improved.

not applicable

Author Response

Dear Reviewer,

We thank the reviewers for their constructive comments. We have addressed the referees’ comments and modified the paper accordingly. 

Numbering was corrected, numbers in pictures lifted above and Figure 5 improved to be more visual.

Thanks

Reviewer 5 Report

Comments to the authors

The present paper aims at enhancing production efficiency in manufacturing companies in order to reduce energy consumption through time savings.

The paper is very well written and structured, and it is of interest for the scientific community. Thus, the reviewer considers that it should be accepted for publication after minor amendments described as follows. The reviewer advises to include a “State-of-the-art” section that will serve to highlight the novelty, short one. Besides, the paper requires some minor amendments, for which detailed comments are provided as follows.

·         In the abstract, the authors claim a 5% improvement. Could the authors justify (maybe comparing with other studies) if this improvement is significant?

·         The Introduction describes briefly the problem of energy consumption for manufacturing companies and introduce the method implemented in the present study (Description of Decision Support Method for Dynamic Production Planning). The reviewer is missing a State-of-the-art Section, not too long, including different existing methods with similar objectives. In the reviewer’s opinion, this section is of high relevance in order to highlight the novelty of the present work and, thus, the contribution to the knowledge.

·         Line 64: Once defined the acronym for small and medium enterprises (SME) the full term should not be used any more.

·         Line 68: “but changes sequence of” à but changes THE sequence of.

·         Excessive use of the word “investigation”, e.g. lines 73 – 75, more examples in the text.

·         Please revise captions in figures and add missing final dots (in some).

·         Line 105: “This section presents general methodology” à This section presents THE general methodology.

·         Figure 2: the font size of the text boxes is too small and hard to read. Please could the authors maybe try to increase it? It is a great diagram.

·         Line 136 a coma is missing: “which is needed should” à “which is needed, should”.

·         Table 1: without horizontal lines it is hard to visualize where the list products finishes and the list of services begin. Although this can be deduced, it is probably better too make it clear introducing horizontal lines. Besides, the listed products and services should not have “;” or “.” after each word (because it is a table).

·         Lines 201 & 204: those “it was agreed” to stablish 5% and 25% of the total electricity are based on what assumptions?

·         Lines 218 to 223: please add final dot at the end of each element.

·         Table 3: normally units are indicated using squared brackets [], thus, it should read “Active power [kW]”, “Standby power [kW]”.

·         Figure 5 appears after figures 6 to 9, which complicates the reading. The reviewer believes that figure 5 could appear in page 9 so before the mentioned subsequent figures.

·         The outcomes obtained for figures 6 to 11 are not explained, please comment at least the main observations.

·         Figures 21, 22 and 23 are not high quality (as the rest), please amend.

·         Just an opinion, which can be disregarded, the reviewer considers that figures 6 to 22 should not have an outer frame. Specially in figures 6 to 20 because it appears to have been cut which only show 3 (out of the 4) parts of a frame. For consistency, they should all look the same.

Author Response

We thank the reviewers for their constructive comments. We have addressed the referees’ comments and modified the paper accordingly. Our detailed answers follow. Please note that the reviewers' comments are in bold while our answers are not. Furthermore, additions to the revised manuscript are indicated in red.

Itemized reply to reviewer 5

Dear reviewer,

Thank you for such a constructive review.

Below, we answer your comments one by one:

  • The reviewer advises to include a “State-of-the-art” section that will serve to highlight the novelty, short one.

Thanks for this recommendation. We have included a Literature review section with comments about novelty of our method.

  • In the abstract, the authors claim a 5% improvement. Could the authors justify (maybe comparing with other studies) if this improvement is significant?

Thanks for the observation. We have included information about the influence of 5% to the abstract.

  • The Introduction describes briefly the problem of energy consumption for manufacturing companies and introduce the method implemented in the present study (Description of Decision Support Method for Dynamic Production Planning). The reviewer is missing a State-of-the-art Section, not too long, including different existing methods with similar objectives. In the reviewer’s opinion, this section is of high relevance in order to highlight the novelty of the present work and, thus, the contribution to the knowledge.

We completely agree with the reviewer about the importance of this section. Hence, we have included a “Literature review” section, with comments about novelty of our method.

  • Line 64: Once defined the acronym for small and medium enterprises (SME) the full term should not be used any more.

Thanks for the advice. We have corrected it.

  • Line 68: “but changes sequence of” à but changes THE sequence of.

Thank you. We have corrected it.

  • Excessive use of the word “investigation”, e.g. lines 73 – 75, more examples in the text.

We appreciate the recommendation and have corrected in the whole text.

  • Please revise captions in figures and add missing final dots (in some).

Thanks for the comment. We have revised all figure captions and added missing final dots in the required ones.

  • Line 105: “This section presents general methodology” à This section presents THE general methodology.

Thank you. We have corrected it.

  • Figure 2: the font size of the text boxes is too small and hard to read. Please could the authors maybe try to increase it? It is a great diagram.

Thank you. We have corrected it.

  • Line 136 a coma is missing: “which is needed should” à “which is needed, should”.

Thank you. We have corrected it.

  • Table 1: without horizontal lines it is hard to visualize where the list products finishes and the list of services begin. Although this can be deduced, it is probably better to make it clear introducing horizontal lines. Besides, the listed products and services should not have “;” or “.” after each word (because it is a table).

We appreciate this recommendation and have implemented in the revised manuscript (now as Table 2).

  • Lines 201 & 204: those “it was agreed” to stablish 5% and 25% of the total electricity are based on what assumptions?

Thanks for this comment. We have explained more in the revised manuscript why this was selected.

  • Lines 218 to 223: please add final dot at the end of each element.

Thank you. We have added final dots.

  • Table 3: normally units are indicated using squared brackets [], thus, it should read “Active power [kW]”, “Standby power [kW]”.

Thanks for this recommendation. We have corrected in new table 4 of the revised manuscript.

  • Figure 5 appears after figures 6 to 9, which complicates the reading. The reviewer believes that figure 5 could appear in page 9 so before the mentioned subsequent figures.

Sorry, we didn’t realize about this mistake. It is fixed now. Thanks.

  • The outcomes obtained for figures 6 to 11 are not explained, please comment at least the main observations.

We appreciate this observation. Hence, we have included the main observations of these figures before all the images.

  • Figures 21, 22 and 23 are not high quality (as the rest), please amend.

Thank you. We have corrected it.

  • Just an opinion, which can be disregarded, the reviewer considers that figures 6 to 22 should not have an outer frame. Specially in figures 6 to 20 because it appears to have been cut which only show 3 (out of the 4) parts of a frame. For consistency, they should all look the same.

Thank you, we have followed your recommendation and the figures look better now.

Round 2

Reviewer 1 Report

The authors have addressed all my concerns.

Reviewer 3 Report

Thank you for taking into account the reviewers' suggestions. In my opinion, the article can be published in its current form.